# MOCVD of II-VI HRT/Emitters for $V_{oc}$ Improvements to CdTe Solar Cells

Andrew J. Clayton [1], Ali Abbas [2], Peter J. Siderfin [1], Stephen Jones [1], Ana Teloeken [1], Ochai Oklobia [1], John M. Walls [2] and Stuart J. C. Irvine [1,*]

1 Centre for Solar Energy Research, Swansea University, OpTIC Centre, Ffordd William Morgan, St. Asaph Business Park, Denbighshire LL17 0JD, UK; Andrew.J.Clayton@Swansea.ac.uk (A.J.C.); p.j.siderfin@swansea.ac.uk (P.J.S.); stephen.jones@swansea.ac.uk (S.J.); acteloeken@hotmail.com (A.T.); Ochai.Oklobia@swansea.ac.uk (O.O.)
2 Centre for Renewable Energy Systems Technology, Wolfson School of Mechanical, Electrical and Manufacturing Engineering, Loughborough University, Loughborough, Leicestershire LE11 3TU, UK; a.abbas@lboro.ac.uk (A.A.); J.M.Walls@lboro.ac.uk (J.M.W.)
* Correspondence: S.J.C.Irvine@Swansea.ac.uk; Tel.: +44-1745-535-234

**Abstract:** CdTe solar cells were produced using metal organic chemical vapour deposition (MOCVD), which employed a (Zn,Al)S (AZS) high resistant transparent (HRT) layer at the transparent conducting oxide (TCO)/Cd(Zn)S emitter interface, to enable the higher annealing temperature of 440 °C to be employed in the chlorine heat treatment (CHT) process. The AZS HRT remained intact with conformal coverage over the TCO after performing the high CHT annealing, confirmed by cross-section scanning transmission electron microscopy coupled with energy-dispersive X-ray spectroscopy (STEM-EDX) characterisation, which also revealed the Cd(Zn)S emitter layer having been consumed by the CdTe absorber via interdiffusion. The more aggressive CHT resulted in large CdTe grains. The combination of AZS HRT and aggressive CHT increased open circuit voltage ($V_{oc}$) and improved solar cell performance.

**Keywords:** MOCVD; CdTe; photovoltaics; STEM-EDX

## 1. Introduction

The rapid rise in photovoltaic (PV) solar energy over the past decade has been largely driven by the supply of crystalline silicon PV modules. The total installed PV capacity in 2019 exceeded 600 GW with additional capacity of over 100 GW added in 2020 alone [1]. Alongside the exponential growth of silicon PV, there has been a similar growth in thin film PV, but this started from a much lower base which has left thin film PV with a total market share of less than 10% [2]. This now mainly comprises thin film cadmium telluride (CdTe) and copper indium gallium diselenide (CIGS) with the larger share attributed to CdTe. Progressive improvements in the solar conversion efficiency of thin film PV have narrowed the gap with crystalline silicon, which has a world best cell efficiency of 26.7%, with current world record CdTe solar cells having 22.1% efficiency under standard air mass conditions (AM1.5) [3]. Despite this impressive progress, there is still a way to go with a theoretical limit for single junction CdTe cells of around 30% and a practical prospect for >25% efficiency [4]. The next step in CdTe cell efficiency will depend on changing the acceptor doping from copper to arsenic which has demonstrated improvement by more than a factor of 10 in acceptor concentration [5–9]. Some of this pioneering work on increasing the acceptor concentration has used metal organic chemical vapour deposition (MOCVD) to establish doping mechanisms that have then been replicated by more commonly used sublimation and physical vapour transport techniques [9,10]. To realise the benefits of higher acceptor doping concentrations, it will become necessary to develop different emitters than the currently used CdS, $SnO_2$ and (Mg,Zn)O. The emitter needs to

have a slight positive conduction band offset with the absorber, to minimise interfacial recombination and donor doping at a concentration significantly higher than the acceptor concentration in the absorber layer [4]. The latter is to ensure that the depletion region is largely in the absorber layer, away from the defective hetero-interface. For CdTe solar cells, the substrate (referred to as a superstrate) becomes the window to admit solar radiation so the front contact is a transparent conducting oxide (TCO) such as fluorine-doped tin oxide (FTO) [11,12]. This is sometimes buffered with a high-resistance transparent layer (HRT) to avoid shunting through to the CdTe layer to the back contact. The issue here is that the FTO has a rough, faceted surface, so achieving a conformal coverage of a 30–50 nm thick HRT layer requires excellent nucleation and growth of the film. This has enabled thinner CdS emitter layers resulting in less parasitic absorption at wavelength <500 nm [13].

It has previously been shown that an (Al,Zn)S (AZS) layer, grown by MOCVD, can enable a thinner (Cd,Zn)S emitter layer without sacrificing the key performance parameters of fill factor (*FF*) and open circuit voltage ($V_{oc}$) [14]. It was also found that the Al doping made the AZS film more chemically resilient. This enabled a higher temperature, post-growth, chlorine heat treatment (CHT) that is used to achieve larger CdTe grains with more randomised texture. The chlorine complex formation with impurities that reside at grain boundaries resulting in defect passivation also became more efficient. These benefits require the AZS layer to have extremely good conformal coverage of the rough FTO transparent conductor and to survive the aggressive (agg.) CHT. The authors reported [14] the low electron affinity of AZS creates a positive conduction band discontinuity (CBD) by correlating experimental current-density—voltage (*J-V*) curves with solar cell capacitance simulations (SCAPS) [15]. Both experimental and SCAPS *J-V* curves showed a second rectification when the Cd(Zn)S emitter layer was omitted from the sample to produce an AZS/CdTe interface. Solar cell performances were not as high without the Cd(Zn)S emitter, but the *J-V* curves suggested that after post-growth CHT, an AZS layer remained in the device structure and warranted further investigation.

Scanning transmission electron microscope (STEM) images of cross-sections through the FTO/AZS/CdZnS/CdTe layer structure are used to study nucleation and growth of the MOCVD grown layers and subsequent changes resulting from different CHT conditions [16,17]. An in-line MOCVD process was used to deposit the AZS films over FTO/glass superstrates 10 cm$^2$ in area. The vapour pressures of the chemical precursors were an order of magnitude greater than similar processes for the conventional horizontal MOCVD process [18]. Injection of the precursors was directly over the superstrate at an angle of 60 degrees off-normal to the superstrate surface. This resulted in a high supersaturation of precursor vapour over the superstrate surface creating optimum conditions for high nucleation density across the faceted polycrystalline FTO layer [17]. STEM, coupled with energy-dispersive X-ray spectroscopy (STEM-EDX), was employed to investigate the device structure and elemental compositions by characterising cross-sections of solar cells having undergone different CHT conditions, and the findings are provided in this report.

## 2. Materials and Methods

The AZS films were deposited using an in-line MOCVD system using diethylzinc (DEZn), ditertiarybutyl sulphide (DtBS) and trimethylaluminium (TMAl) as metalorganic precursors for Zn, S and Al, respectively. A nitrogen carrier gas was used to transport the chemical vapour at a total flow of 250 standard cubic centimetres per minute (sccm) with precursor partial pressures (atmospheres) of $3.7 \times 10^{-3}$ atm for DtBS, $1.8 \times 10^{-3}$ atm for DEZn and $9.2 \times 10^{-4}$ atm for TMAl, which represented an injection ratio of 4:2:1 for S:Zn:Al. The substrate was NSG TEC10 with area of 10 cm$^2$, which had FTO with sheet resistance of 10 ohms per square (ohms/sq). The surface temperature was 400 °C and was placed on a linear stage which moved at a speed of 5.4 cm/min during deposition.

The subsequent CdS/Cd(Zn)S/CdTe:As layer depositions and CHT process were carried out on a conventional horizontally configured MOCVD reactor, with a hydrogen carrier gas and growth temperatures of 315 °C for the CdS nucleation layer using dimethyl

cadmium (DMCd) and DtBS, 360 °C for the Cd(Zn)S emitter using DMCd, DEZn and DtBS and 390 °C for the arsenic-doped CdTe:As absorber layer using DMCd, diisopropyltelluride (DiPTe) and trisdimethylaminoarsenic (TDMAAs). The CHT was carried out by depositing CdCl$_2$ at 200 °C using DMCd and tertiarybutylchloride (t-BuCl), followed by an anneal at standard (std.) temperature of 420 °C for 10 min, or using an aggressive (agg.) anneal at 440 °C for 10 min. Devices that did not have the CHT process were completed without the CdCl$_2$ layer and anneal. All devices received a final anneal in air at 170 °C for 90 min [17]. Gold (Au) contacts with area of 0.25 cm$^2$ were deposited under vacuum using a thermal evaporator. Front contacts were established by removing the layers around the edge of the device down to the FTO and silver (Ag) paint was applied to increase conductivity.

STEM (Philips, Eindhoven, The Netherlands) was carried out using an FEI Tecnai F20 instrument. Bright and dark field images were collected using a Gatan STEM detector. Coupled to the STEM was an Oxford Instruments X-Max 80 mm$^2$ windowless EDX spectrometer. A silicon drift detector (Oxford Instruments, Oxford, UK) was used with the STEM instrument for high spatial resolution EDX elemental mapping. Current-density—Voltage (*J-V*) measurements were carried out using an Abet Technologies Ltd. solar simulator (Milford, CT, USA) with light density output of 100 mW/cm$^2$ at air mass (AM1.5) and calibrated using a shunted RR-1004 Rera Solutions GaAs reference cell. The voltage sweep was carried out using a Keithly 2400 source meter with *J-V* parameters recorded using in-house programming with Visual Basic software (Microsoft Visual Basic 6.0).

## 3. Results and Discussion

All the device samples consisted of a 3 micron (μm) CdTe absorber layer with 100 nanometre (nm) Cd(Zn)S emitter layer deposited on to a 50 nm CdS nucleation layer. The Zn(Al)S (AZS) layer was either 50 nm or 100 nm, as detailed further in this report, and was deposited as the first layer on NSG TEC10 substrates, which consisted of fluorine-doped tin oxide (FTO) on soda-lime glass produced by NSG Pilkington. The device structure is represented in Figure 1a, with energy band diagrams for device stacks with Figure 1b and without Figure 1c the AZS layer.

The CdTe absorber layer was doped with arsenic (As) to give p-type conductivity with acceptor concentrations ($N_a$) of $1 \times 10^{16}$ cm$^{-3}$ as reported in earlier work [6]. CdTe solar cells are stable in air but can suffer from issues associated with the acceptor dopant, which is typically copper (Cu). It has the tendency to diffuse towards the emitter and create micro shorts. Doping the CdTe absorber with As for the acceptor dopant improves solar cell stability whilst maintaining high solar conversion efficiencies [9].

The benefits obtained by the final CHT of thin film CdTe solar cells are crucial to high PV performance through grain boundary passivation and grain enlargement [19]. The thin film CdTe solar cell structure needs to endure a high-temperature anneal with chlorine diffusing through the CdTe layer from the back surface via the grain boundaries. Grain enlargement is essential for reducing grain boundary density, where high rates of minority carrier recombination can occur. This can be observed in Figure 2, which shows cross-sectional STEM images for a device where CHT was carried out at 420 °C and another with CHT at 440 °C. The increase in grain size is significant, which extends from the back CdTe surface to the absorber/emitter interface in the case of the latter. This correlates with previous work [14] showing cross-sectional scanning electron microscope (SEM) images. Each time the CHT anneal temperature was increased, the grains became more columnar with reduced density of smaller grains at the emitter interface, which correlates with previous reports [20,21]. The CdTe grain for the sample annealed at 420 °C varied from small 0.4 micron (μm) wide grains residing close to the emitter interface, to wider 0.6–1.0 μm grains towards the back CdTe surface from which chlorine is diffused during the CHT anneal. The CdTe grains for the sample annealed at 440 °C became consistently larger at 1.0–1.5 μm and extended further through the absorber layer to 2 μm in length in some areas.

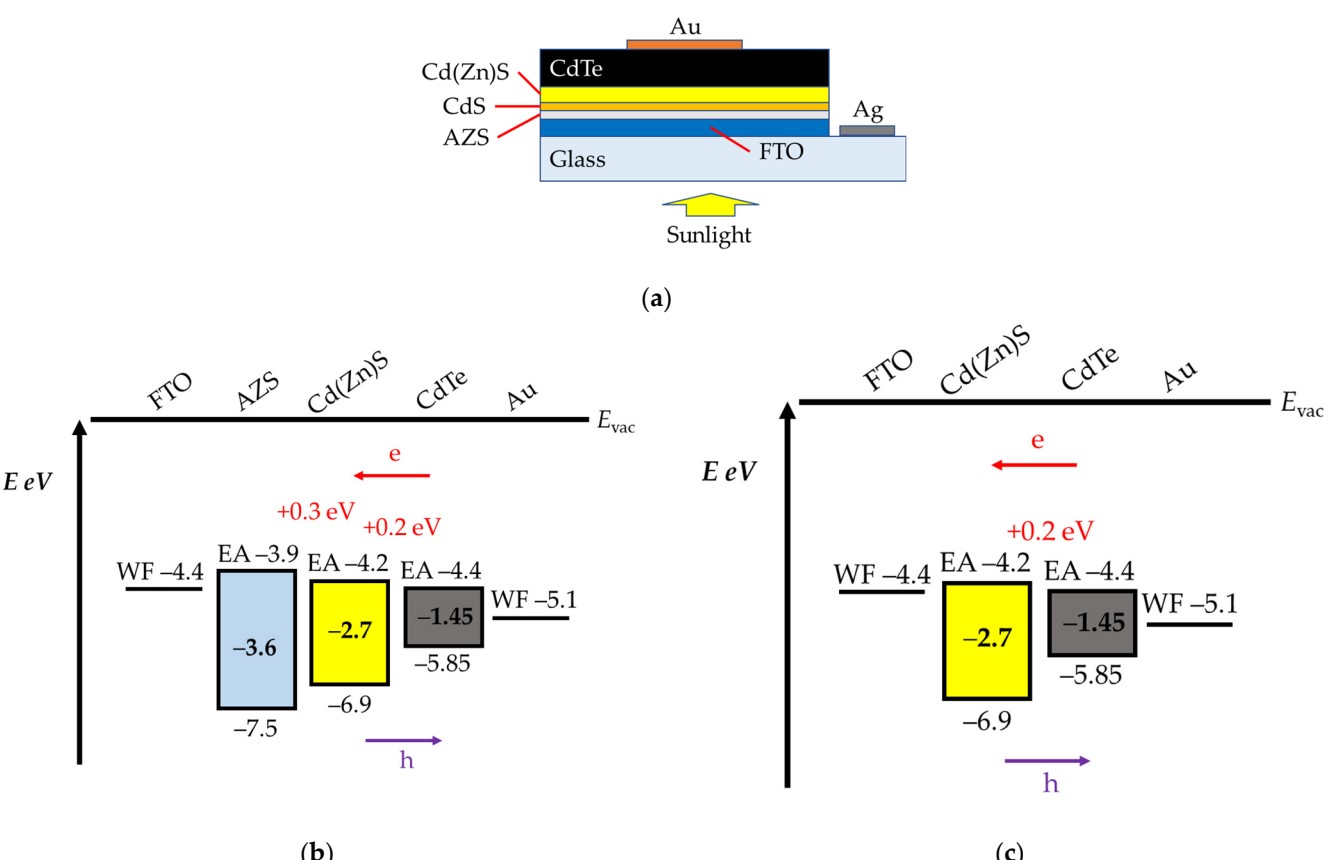

(a)

(b)

(c)

**Figure 1.** Schematic of (**a**) the CdTe device stack for each sample produced in the study; (**b**) energy bands of FTO/AZS/Cd(Zn)S/CdTe:As/Au device stack showing work function (WF) and electron affinity (EA); (**c**) energy bands of FTO/Cd(Zn)S/CdTe:As/Au device stack showing reduced conduction band discontinuity (CBD) with AZS omitted from the structure.

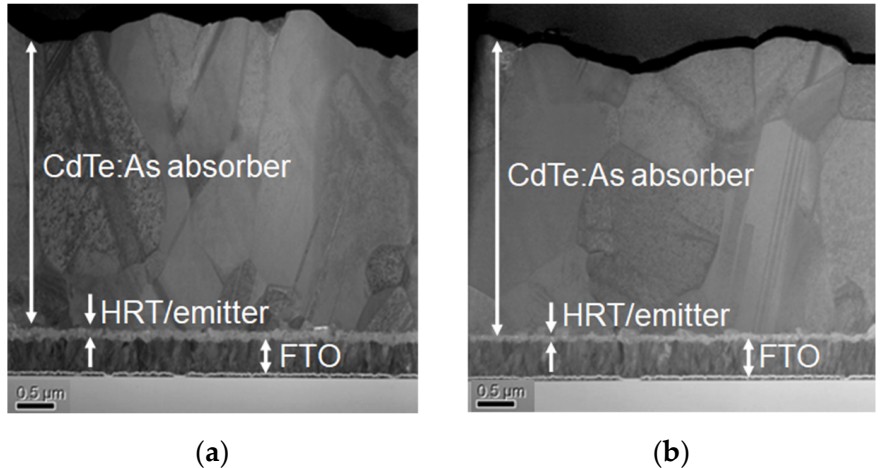

(a)

(b)

**Figure 2.** Cross-sectional STEM of (**a**) CdTe device with CHT at 420 °C and (**b**) CdTe device with CHT at 440 °C.

A previous report [14] showed that an AZS HRT was more resistive to HCl etching than ZnO. The AZS HRT thus allowed more agg. CHT to be employed on the CdTe solar cells with improvements to $V_{oc}$. Without the AZS layer there was significant delamination at the TCO/emitter interface, which was notably observed from the glass side of the sample with the presence of multiple patterns forming with the black CdTe and interdiffused layers coming away from the FTO/glass as previously reported [14]. The CHT results in changes

to the solar cell structure, which include interdiffusion between the emitter and the CdTe absorber layers [16,17].

The presence of the AZS HRT has had a beneficial effect on the $V_{oc}$ and has acted as a chemical barrier during the agg. CHT process. *J-V* parameters (Table 1) demonstrate a correlation between the AZS HRT and CHT process, with higher annealing temperatures improving *FF*, alongside lower series resistance ($R_s$). *J-V* curves (Figure 3) show the improvement to $V_{oc}$ with inclusion of the AZS HRT layer and the increase in CHT anneal temperature. Comparison is made to a baseline device sample with no AZS HRT layer having received a std. CHT process.

**Table 1.** *J-V* parameters with standard deviation of FTO/AZS/CdS/Cd(Zn)S/CdTe:As solar cell devices with different degrees of CHT processing; comparison to a baseline solar cell device with AZS layer omitted.

| Sample | AZS (nm) | CdS (nm) | Cd(Zn)S (nm) | CdTe:As (nm) | CHT (°C) | T °C (min) | No. Cells | PCE (%) | $J_{sc}$ (mA/cm²) | $V_{oc}$ (mV) | FF (%) |
|---|---|---|---|---|---|---|---|---|---|---|---|
| Baseline | - | 50 | 100 | 3000 | 420 | 10 | 8 | 14.1 ± 0.2 | 24.3 ± 0.4 | 760 ± 0 | 76.5 ± 2.1 |
| a | 50 | 50 | 100 | 3000 | - | - | 3 | 1.4 ± 0.4 | 9.4 ± 1.6 | 500 ± 10 | 28.9 ± 3.0 |
| b | 50 | 50 | 100 | 3000 | 420 | 10 | 3 | 6.3 ± 2.7 | 23.1 ± 3.5 | 610 ± 10 | 38.3 ± 11.7 |
| c | 50 | 50 | 100 | 3000 | 440 | 10 | 8 | 9.7 ± 1.6 | 25.5 ± 0.8 | 700 ± 10 | 54.3 ± 8.0 |
| d | 100 | 50 | 100 | 3000 | 440 | 10 | 12 | 14.3 ± 0.4 | 24.1 ± 0.6 | 824 ± 10 | 71.8 ± 1.7 |

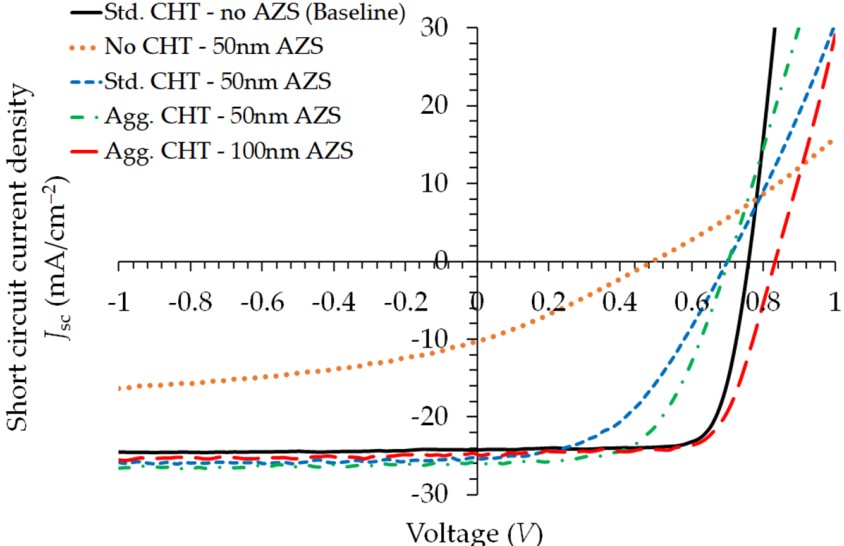

**Figure 3.** *J-V* curves of FTO/AZS/CdS/Cd(Zn)S/CdTe:As solar cell devices with different CHT anneal temperatures, showing comparison to a baseline FTO/CdS/Cd(Zn)S/CdTe:As solar cell device with no AZS HRT and std. CHT processing.

A thicker 100 nm AZS resulted in higher $V_{oc}$ relative to the device having the same CHT process but with a 50 nm AZS HRT. The STEM-EDX for both these devices (Figure 4) showed a strong presence of ZnS at the TCO interface. The AZS HRT layer remaining at the CdTe interface provided a positive CBD, which was a contributing factor to the increased $V_{oc}$ and overall improvement in *J-V* parameters in a previous report [14], showing the energy band diagram for equivalent CdTe device stacks. This was attributed to the low electron affinity of AZS of 3.9 eV and the positive CBD of +0.5 eV when interfacing with CdTe, which has electron affinity equal to 4.4 eV. AZS was even shown to act as an emitter in the absence of Cd(Zn)S, although *J-V* curves were susceptible to a second rectification. The presence of the Cd(Zn)S emitter prior to CHT reduced this CBD, which has an electron affinity of 4.2 eV to create a smaller +0.2 eV band offset for Cd(Zn)S/CdTe,

Figure 1b,c, compared to the +0.5 eV band offset at the AZS/CdTe interface. However, the AZS/Cd(Zn)S interface introduces another +0.3 eV band offset (Figure 1b) creating another barrier for minority carrier collection. The agg. CHT also resulted in larger grains, another contributing factor for increased $V_{oc}$ with less minority carrier recombination at the grain boundaries.

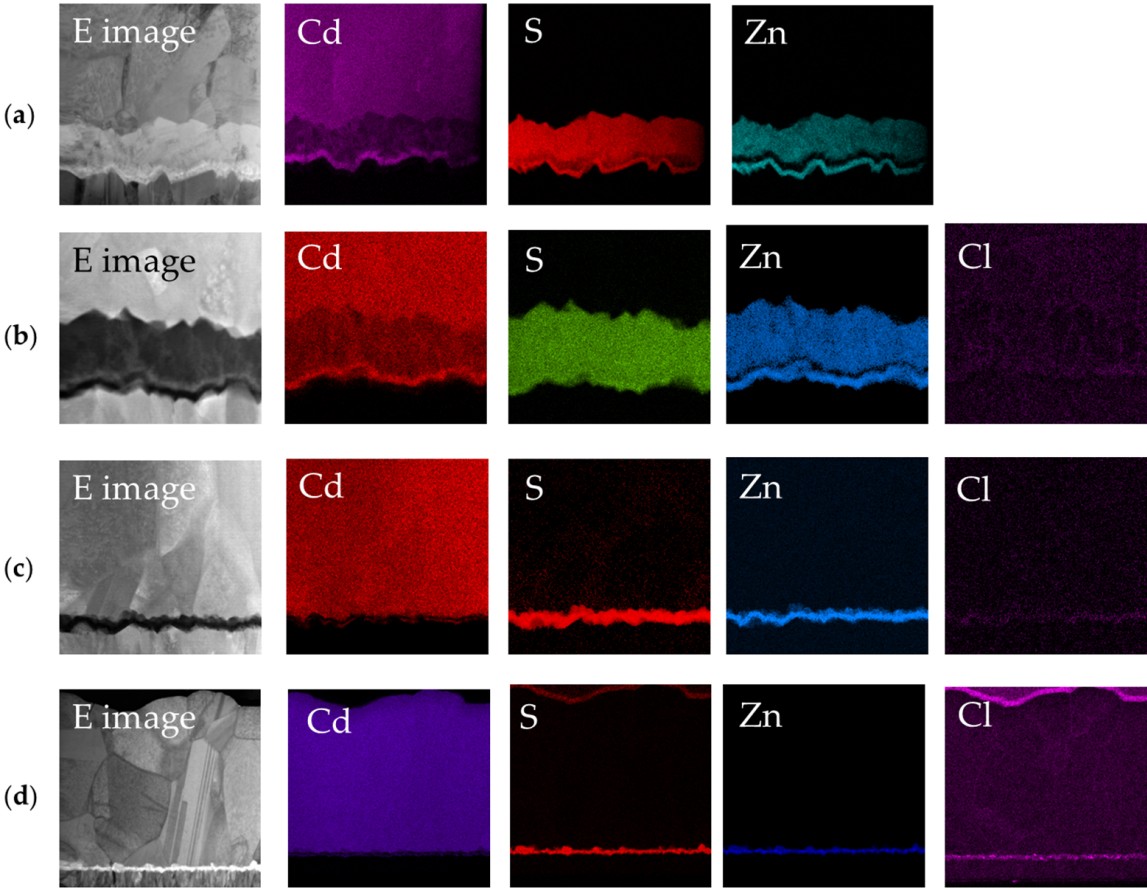

**Figure 4.** Cross section STEM image with elemental EDX maps for Cd, S, Zn and Cl in FTO/AZS/CdS/Cd(Zn)S/CdTe:As solar cell device cross-sections taken alongside STEM images with different degrees of CHT processing; (**a**) device sample receiving no CHT, (**b**) device sample with std. CHT at 420 °C, (**c**) device sample with agg. CHT at 440 °C, and (**d**) device sample with thicker 100 nm AZS HRT with agg. CHT at 440 °C.

Figure 4 shows EDX elemental maps of the AZS/CdS/Cd(Zn)S/CdTe device cross-sections having undergone different levels of CHT. The images show different levels of interdiffusion of the Cd(Zn)S emitter and the impact on the HRT/emitter layer structure, which in this case consisted of a 50 nm AZS HRT with 150 nm combined CdS/Cd(Zn)S emitter layer. Previous work [22,23] showed that the Cd(Zn)S emitter composition prior to the CHT process was $Cd_{0.3}Zn_{0.7}S$, which changes to be closer to $Cd_{0.4}Zn_{0.6}S$ after the CHT process via interdiffusion, with the CdS nucleation layer intermixing with the Cd(Zn)S emitter [22]. The images of the untreated device sample (a) show a bright defined layer above the FTO layer (dark region) which corresponds to a ZnS layer, as can be seen from the EDX images. The concentration of Al was below the detectable limit of the EDX spectrometer, which had spatial resolution < 5 nm. It may also have been the case that the Al, being highly mobile, had diffused out of the device structure during the CHT process. Above the ZnS layer, a narrow dark layer (Zn profile) can be seen, corresponding to the CdS layer which is clearly shown in the Cd and S profiles. The Cd(Zn)S layer is revealed in

all three Cd, S and Zn EDX images. The CdS and Cd(Zn)S layers together form the emitter in this device.

After std. CHT (420 °C) for device sample (b), the HRT/emitter profile broadens due to interdiffusion, with less-defined layers in the EDX images although the CdS layer can still be seen by comparing the Cd, S and Zn EDX images. When the CHT becomes more aggressive (sample (c)), the HRT/emitter region narrows as it becomes substantially interdiffused with the CdTe absorber. The only remaining layer that is still visible in the EDX images is the ZnS HRT layer, confirming the robustness of this layer to agg. CHT.

STEM-EDX of device sample (d), which had a 100 nm AZS HRT layer and was subjected to a CHT at 440 °C, shows that after the agg. post-growth treatment, interdiffusion has driven some S from the CdS/Cd(Zn)S emitter all the way to the back CdTe surface (S profile, sample (d)). This has been observed for devices with an ultra-thin 0.5 µm thick CdTe absorber [22] using std. 420 °C CHT conditions. The Cl profile reveals that it has diffused though the whole device structure with some remnant of the deposited $CdCl_2$ layer on the surface. Accumulation can be seen at the TCO interface.

The extent of interdiffusion of the HRT/emitter region is further illustrated with STEM-EDX line scans through the device structure. Figure 5 compares the untreated device (a), shown in Figure 5a, with sample (d), shown in Figure 5b, having had agg. CHT at 440 °C.

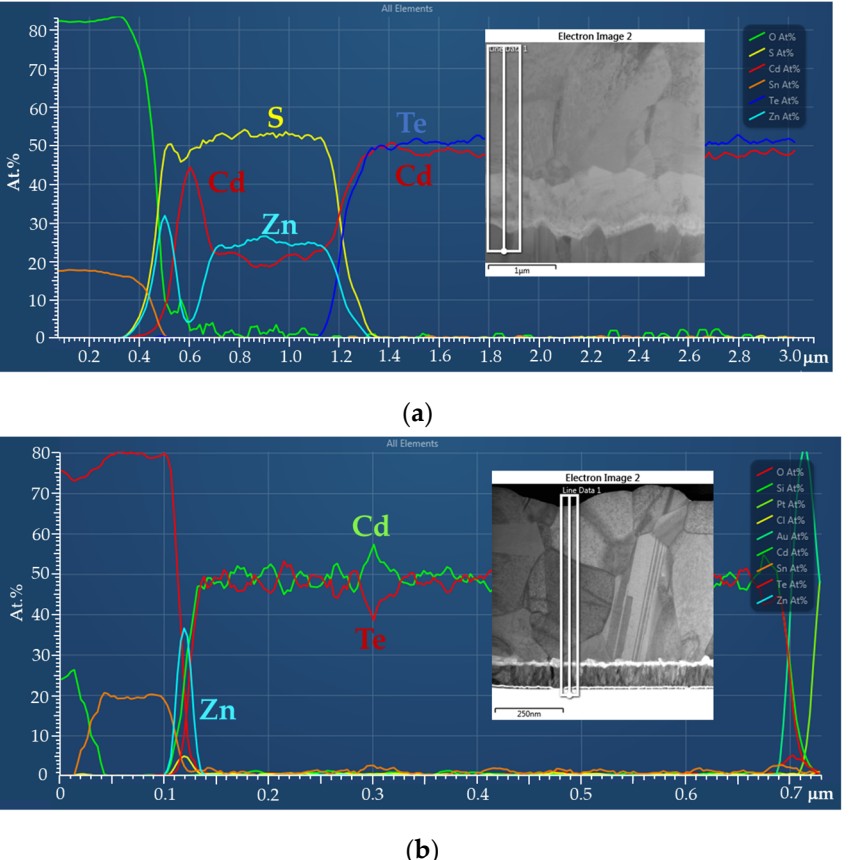

**Figure 5.** Elemental EDX line scans through the device stack of (**a**) FTO/AZS/CdS/Cd(Zn)S/CdTe:As solar cell device with no CHT process; and (**b**) equivalent device structure having undergone aggressive CHT.

The untreated CdTe device sample in Figure 5a clearly shows the defined layers of the HRT/emitter structure from the FTO surface with ZnS/CdS/Cd(Zn)S prior to any interdiffusion [22,23]. No Al in the AZS is again detectable due to dopant concentration being less than 1 At.% and below the detectable limit of the instrument. Figure 5b shows the HRT/emitter after agg. CHT, which narrows significantly after interdiffusion and

shows a prominent Zn peak at the TCO interface. Despite the interdiffusion of the emitter, Figures 4 and 5 show that a ZnS layer remained intact and maintained an interface layer with the FTO.

To further understand the role that the AZS HRT layer had on $V_{oc}$, dark *J-V* curves were used to calculate the reverse saturation current-density ($J_0$) and ideality factor ($n$), which are related to $V_{oc}$ in the following equation:

$$V_{oc} = \frac{nKT}{q} ln\left(\frac{J_l}{J_0} + 1\right) \tag{1}$$

The thermal voltage is represented by $KT/q$ and an ideal diode would have $n = 1$ with only radiative recombination contribution. When minority carrier recombination also occurs, the value of $n$ increases, with typical value between 1 and 2. $J_l$ is the light-generated current-density and can be assumed to be equivalent to $J_{sc}$. Plots were made (Figure 6) of the dark *J-V* curves with ln ($J_D$) over a voltage ($V$) range 0.55–0.7 V.

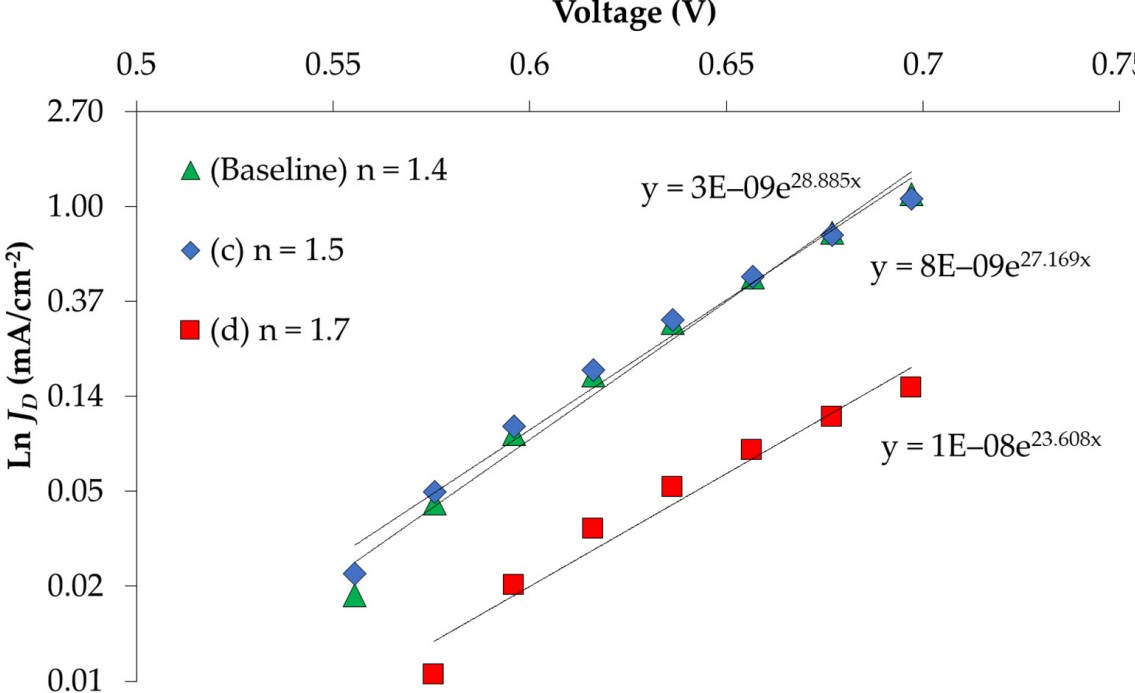

**Figure 6.** Plot of ln ($J_D$-$V$) curve to extract the reverse saturation current-density ($J_0$) and ideality factor ($n$).

The straight lines from the ln ($J_D$-$V$) plot for samples (c) and (d), which had AZS thicknesses of 50 nm and 100 nm, respectively, were used to generate the $J_0$ with gradient of $q/nKT$, which can be represented by α. The value of n was extracted from the gradient using the following equation:

$$n = \frac{q/KT}{\alpha} \tag{2}$$

A baseline device sample, which had no AZS HRT layer, was also plotted with device samples (c) and (d) for comparison. Figure 6 is representative of the data for all cells measured for each device sample. The average $J_0$ and n extracted from the dark *J-V* curves are shown in Table 2. In terms of the diode quality, n is highest for device sample (d), improving for sample (c) and being closest to an ideal diode for the baseline, which has no AZS HRT layer. This does not reflect the light-generated *J-V* data, in which the $V_{oc}$ was greatest for sample (d). The average $J_0$ data were similar between samples (c) and (d), but were an order of magnitude better for the baseline device sample.

**Table 2.** Average reverse saturation current-density ($J_0$) and ideality factor ($n$) extrapolated from dark *J-V* curves for device samples (c) and (d) with comparison to an equivalent device (Baseline) without an AZS HRT layer.

| Sample | AZS (nm) | CdS (nm) | Cd(Zn)S (nm) | CdTe:As (nm) | CHT (°C) | T °C (min) | No. Cells | Mean $n$ | Mean $J_0$ (mA/cm$^2$) |
|--------|----------|----------|--------------|--------------|----------|------------|-----------|----------|------------------------|
| Baseline | - | 50 | 100 | 3000 | 420 | 10 | 8 | $1.4 \pm 0.1$ | $7.2 \times 10^{-9}$ |
| c | 50 | 50 | 100 | 3000 | 440 | 10 | 8 | $1.5 \pm 0.2$ | $2.4 \times 10^{-7}$ |
| d | 100 | 50 | 100 | 3000 | 440 | 10 | 12 | $1.7 \pm 0.0$ | $9.2 \times 10^{-8}$ |

The data in Table 2 suggest that the device structure produces a poorer diode when the AZS HRT is included, which should lead to greater minority carrier recombination and a reduced $V_{oc}$. The high energy barrier in the conduction band caused by the AZS may be detrimental under dark conditions. However, under light conditions, the generated carriers may be saturated at the absorber/emitter interface, driving the drift across this energy spike into the depletion region, where carriers can then be collected. The energy spike in the conduction band creates a larger energy difference between the electron-hole pairs generated under the light conditions, keeping the n-type and p-type carriers separate and reducing recombination. This would lead to the higher $V_{oc}$ observed in the *J-V* performance for device samples with an AZS HRT. Without the energy spike in the conduction band, the difference in energy of electron-hole pairs is smaller, which may lead to more recombination and a lower $V_{oc}$ for the baseline device sample. This may explain the improvement in $V_{oc}$ under light conditions when an AZS HRT is included in the device, although further investigation would be required.

## 4. Conclusions

STEM-EDX showed that an agg. CHT process at 440 °C was essential for achieving large CdTe grains and complete consumption of the emitter to leave the AZS HRT and produce a defined HRT/absorber interface region. EDX profiles of a device with agg. CHT showed that the S profile reached the back CdTe surface, with Cl diffusing down to the TCO, where it accumulated. The higher annealing temperature during the CHT process resulted in an increase in $V_{oc}$ relative to a standard CHT process at 420 °C. An AZS HRT layer 100 nm thick improved $V_{oc}$ relative to using a 50 nm AZS layer. The large positive CBD in the conduction band with the inclusion of the AZS HRT increases the energy difference of electron-hole pairs to reduce recombination and improve $V_{oc}$. Investigation into the dark *J-V* characteristics for device samples (c) and (d), both having undergone agg. CHT post growth, showed that the inclusion of an AZS HRT layer increased the reverse saturation current-density and ideality factor in comparison to a baseline device sample without an AZS HRT layer. This infers that minority recombination was greater for devices with AZS, which did not correlate with the *J-V* parameters under light conditions. The energy difference between electron-hole pairs was greater for the device samples with AZS, reducing the probability of recombination. This resulted in higher $V_{oc}$ for device samples with AZS relative to the baseline device sample (no AZS), where the energy difference between electron-hole pairs was closer, giving more probability of recombination reducing the $V_{oc}$. The energy barrier caused by inclusion of the AZS layer had a detrimental effect on the diode quality. However, the greater energy provided under light conditions led to saturation of generated carriers in this region. This drove the PV drift across the depletion region where the carriers were collected, leading to the observed increase in *J-V* parameters.

**Author Contributions:** Funding acquisition and supervision, S.J.C.I. and J.M.W.; investigation, P.J.S., S.J. and A.T.; methodology, A.J.C., P.J.S., S.J. and O.O.; characterisation, P.J.S., S.J., O.O. and A.A.; data curation, A.J.C.; writing—original draft preparation, S.J.C.I. and A.J.C.; writing—review and editing, all authors. All authors have read and agreed to the published version of the manuscript.

**Funding:** This work was funded by the Europe Regional Development Fund (ERDF) through the Welsh European Funding Office (WEFO) on the 2nd Solar Photovoltaic Academic Research Consortium (SPARC II) project, case number 81133, and the Engineering and Physical Sciences Research Council (EPSRC) funded project EP/W000555/1.

**Institutional Review Board Statement:** Not applicable.

**Informed Consent Statement:** Not applicable.

**Data Availability Statement:** Data used in this publication are archived at DOI:10.5281/zenodo.5784060.

**Conflicts of Interest:** The authors declare no conflict of interest.

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
