# Peer review of "MOCVD of II-VI HRT/Emitters for Voc Improvements to CdTe Solar Cells"

_coatings, doi:10.3390/coatings12020261_

Round 1

Reviewer 1 Report

     Dear authors, I appreciate the idea developed by you in this research and the efforts you made, but there is a lot of room for improvement, so that your research can be expressed according to custom.

    In the text of the article attached to this review I made some remarks, which I will complete in the following.

    A big minus is that the figures were not named correctly (not at all) - figures 3 and 4.

   In Chapter 2, additions should be made to the methods and equipment used for recording J-V curves.

line 123: ``the increase in the grain size is significant`` should be supported by solid statistical data on several areas of the samples, because it is the base of the findings of this research and has no support. Some grain measurements should also appear in Figure 1. 

Because there is no clear explanation of the J-V measurements, figure 2 is not very valuable, so the comments related to these data must be completed. It should also be mentioned on how many samples the tests were performed.

The chapter on conclusions should be extended to support the title of the paper.

Author Response

Dear Review 1,

Please find my response in the attached.

Best regards,

Andrew

Reviewer 2 Report

In this paper, CdTe solar cells were fabricated by utilizing MOCVD technique. These cells have a high resistant transparent layer (AZS) which facilitates a higher temperature annealing resulting in large CdTe grains. The more aggressive CHT produces a higher Voc and improves the fill factor. The work in this paper is interesting but needs some modifications to be considered for possible publication. Here are some comments to improve the work:

  1. A schematic figure for the device structure is needed along with the energy band diagram showing the different energy levels of the various materials used in fabrication.
  2. What is the expected Cd fraction in the ternary compound CdZnS?
  3. The photovoltaic parameters (Jsc, Voc, FF and efficiency) of the presented cells should be given in a table.
  4. The caption of figure 3 is not written correctly. Also, there is no caption for figure 4.
  5. What are the expected doping levels of the layers?
  6. The dark characteristics of the fabricated cells will significantly help understanding the Voc So, it is recommended to draw the dark JV characteristics and to extract the reverse saturation current and ideality factor to give a more physical insight into the experimental results.
  7. The conclusion should be rewritten to include some quantitative results about the photovoltaic parameters.
  8. Overall writing-skill of this paper is good. However, there are some grammatical mistakes, which must be pointed out by the authors and be corrected subsequently.

Author Response

Dear Reviewer 2,

please find my response in the attached.

Best regards,

Andrew

Reviewer 3 Report

The authors employed a (Zn,Al)S (AZS) high resistant transparent (HRT) layer at the (TCO)/Cd(Zn)S emitter interface in the chlorine heat treatment (CHT) process. They suggested Cd(Zn)S emitter layer having been consumed by the CdTe absorber via interdiffusion.  The combination of AZS HRT and aggressive CHT increased Voc and improved solar cell performance. This work showed a systematic study and provides new approach in enhancing device performance. However, there are some concerns as follows:

  1. For the devices, the histogram of device performance and parameters should be presented in order to clearly show the evolution of photovoltaic performance after the introduction of AZS?
  2. Since the AZS plays important roles in the interfaces, does that influence the current-voltage hysteresis of devices?
  3. The stability of devices are preferred to provided.
  4. Has the author investigated the AZS thickness effects on solar cell parameters?
  5. The authors are suggested to check all reference format. 

Author Response

Dear Reviewer 3,

Please find my response in the attached.

Best regards,

Andrew

Round 2

Reviewer 1 Report

I appreciate the changes made according to the requirements of the reviewers. Good luck!

Author Response

The authors are grateful to the reviewer for their comments.

Reviewer 2 Report

The authors have addressed the issues raised by the reviewers. However, I am against the omitting of the J-V figure. It is important to include the illuminated J-V curves as well as the quantitative results. So, both of them are required.

I recommend to include the estimated energy levels of the various materials with respect to vacuum level in Fig. 1. This will give valuable information about the CBO and VBO. 

Author Response

The authors are grateful to the reviewer for their comments and have made the changes requested.
